# Angiotensin-Converting Enzyme 2 (ACE2) in the Context of Respiratory Diseases and Its Importance in Severe Acute Respiratory Syndrome Coronavirus 2 (SARS-CoV-2) Infection

**DOI:** 10.3390/ph14080805

**Published:** 2021-08-17

**Authors:** Enrique Ambrocio-Ortiz, Gloria Pérez-Rubio, Alma D. Del Ángel-Pablo, Ivette Buendía-Roldán, Leslie Chávez-Galán, Rafael de Jesús Hernández-Zenteno, Alejandra Ramírez-Venegas, Jorge Rojas-Serrano, Mayra Mejía, Rogelio Pérez-Padilla, Cristóbal Guadarrama-Pérez, Ramcés Falfán-Valencia

**Affiliations:** 1HLA Laboratory, Instituto Nacional de Enfermedades Respiratorias Ismael Cosío Villegas, Mexico City 14080, Mexico; e_ambrocio@iner.gob.mx (E.A.-O.); gperezrubio@iner.gob.mx (G.P.-R.); alyde_08@hotmail.com (A.D.D.Á.-P.); 2Translational Research Laboratory on Aging and Pulmonary Fibrosis, Instituto Nacional de Enfermedades Respiratorias Ismael Cosio Villegas, Mexico City 14080, Mexico; ivettebu@yahoo.com.mx; 3Laboratory of Integrative Immunology, Instituto Nacional de Enfermedades Respiratorias Ismael Cosio Villegas, Mexico City 14080, Mexico; lchavezgalan@gmail.com; 4Tobacco Smoking and COPD Research Department, Instituto Nacional de Enfermedades Respiratorias Ismael Cosio Villegas, Mexico City 14080, Mexico; rafherzen@yahoo.com.mx (R.d.J.H.-Z.); aleravas@hotmail.com (A.R.-V.); perezpad@gmail.com (R.P.-P.); 5Interstitial Lung Disease and Rheumatology Unit, Instituto Nacional de Enfermedades Respiratorias Ismael Cosío Villegas, Mexico City 14080, Mexico; jrojas@iner.gob.mx (J.R.-S.); medithmejia1965@gmail.com (M.M.); 6Respiratory Emergency Unit, Instituto Nacional de Enfermedades Respiratorias Ismael Cosio Villegas, Mexico City 14080, Mexico; cris.iner@gmail.com

**Keywords:** Angiotensin-Converting Enzyme 2 (ACE2), severe acute respiratory syndrome coronavirus 2 (SARS-CoV-2), renin-angiotensin system (RAS), Coronavirus disease 2019 (COVID-19), angiotensin

## Abstract

Angiotensin-Converting Enzyme 2 (ACE2) is an 805 amino acid protein encoded by the ACE2 gene expressed in various human cells, especially in those located in the epithelia. The primary function of ACE2 is to produce angiotensin (1–7) from angiotensin II (Ang II). The current research has described the importance of ACE2 and Ang (1–7) in alternative routes of the renin-angiotensin system (RAS) that promote the downregulation of fibrosis, inflammation, and oxidative stress processes in a great variety of diseases, such as hypertension, acute lung injury, liver cirrhosis, and kidney abnormalities. Investigations into the recent outbreak of the new severe acute respiratory syndrome coronavirus 2 (SARS-CoV-2) have revealed the importance of ACE2 during infection and its role in recognizing viral binding proteins through interactions with specific amino acids of this enzyme. Additionally, the ACE2 expression in several organs has allowed us to understand the clinical picture related to the infection caused by SARS-CoV-2. This review aims to provide context for the functions and importance of ACE2 with regards to SARS-CoV-2 in the general clinical aspect and its impact on other diseases, especially respiratory diseases.

## 1. Introduction

Angiotensin-Converting Enzyme 2 (ACE2) (EC 3.4.15.1) is an 805 amino acid protein encoded by the ACE2 gene located in cytogenetic band 22.2 on the short arm of the X chromosome. The gene is 41,115 base pairs (bp) long and is organized into 19 exons and 18 introns that can give rise to five different transcripts, of which only two are translated into functional proteins (Figure 1) [1]. In addition, there are two paralogs to ACE2, Angiotensin-Converting Enzyme 1 (ACE) and AC113554.1, although no functional protein product has been associated with the latter.

ACE2 was first described in patients with heart failure [2]; initially, it was suggested that it is expressed in the heart and kidneys. However, recent research has shown that different cells can express ACE2, mainly those located in the epithelium of specific organs, such as the lungs, and fulfill various biological functions, mainly those related to the immune response and homeostasis [3,4,5].

The recent outbreak of the new SARS 2 coronavirus strain (SARS-CoV-2) in late 2019 and its worldwide spread in the first three months of 2020 put ACE2 in the scientific spotlight due to its involvement in the adherence and infection of cells of the pulmonary epithelium. This review aims to provide context for the functions and importance of ACE2 with regards to SARS-CoV-2 in the general clinical aspect and its impact on other diseases, especially respiratory diseases.

## 2. Location and Expression

ACE2 is an enzyme located in the cellular membrane of different human body organs, mainly in the epithelium. Through analyses of expression and proteomic databases, as well as in vitro investigations, it has been shown that ACE2 is expressed in the lungs, liver, kidneys, stomach, intestines, arteries and veins, heart, oral mucosa, nasopharynx, colon, thymus, bladder, and central nervous system [6,7,8,9,10]. The primary cells that express this enzyme are stomach epithelial cells, proximal renal tubes, vascular endothelial cells, glia, neurons, spinal cord cells, cholangiocytes, esophageal keratinocytes, ileal and rectal enterocytes, and epithelial oral cavity epithelial cells [7,8,11,12,13]. The lungs are the main organs of expression at the respiratory system level, with type 2 alveolar and pulmonary epithelial cells as the principal cells in ACE2 expression [11]. The expression level depends on the degree of cell differentiation, and mature cells show the highest level of ACE2 expression (Figure 2) [14]. In animal models, it has been demonstrated that ACE2 expression is decreased in males compared to females, although this difference was believed to be due to estrogens [15].

The research about ACE2 and other potential receptors has been studied in different biological samples. The expression of ACE2 is increased in patients infected with SARS-CoV-2 in the airway epithelium and immunological cells but not in other epithelia [16,17,18]. In some studies, the differences are not significant. In an analysis of the patients infected with SARS-CoV-2, including environmental variables like smoking or other lung diseases (asthma and COPD), the expression of ACE2 is increased significantly, and, interestingly, this further increases in severe COVID-19 [17].

## 3. ACE2/Ang (1–7)/MasR Effector Axis

ACE2 is a member of the counter-regulatory axis of the renin–angiotensin system (RAS), and its leading role is to degrade the pro-hypertrophic and profibrotic peptide angiotensin II (Ang II), limiting the adverse effects of angiotensin II. Additionally, it participates in generating angiotensin (1–7) (Ang (1–7)) (Figure 3) in cooperation with neprilysin, mainly in the liver and kidneys; in the lungs, ACE2 works in conjunction with prolyl oligopeptidase (POP) to generate Ang (1–7) [19]. There are different pathways related to the RAS axis; the most-studied is the ACE axis, Ang II, and the angiotensin II type 1 receptor (AT1R), called the classical system, while ACE2, Ang (1–7), and the Mas receptor (MasR) is one of the nonclassical pathways.

There is an inverse relationship between the ACE/Ang II/AT1R axis and ACE2/Ang (1–7)/MasR; a worsening of the inflammatory processes occurs when this relationship favors the first set, while it favors an increase in the levels of α-smooth muscle actin (α-SMA) and type I collagen at the lung and hepatic levels in fibrotic models [20]. In cellular assays, the phosphorylation levels of vascular endothelial growth factor receptor 2 (VEGFR2), c-Jun, mitogen-activated protein kinase 1/2 (MEK1/2), and extracellular signal-regulated protein kinase (ERK1/2) have been found to decrease via stimulation of the nonclassical pathway of the RAS axis, which significantly inhibits the vascularization processes of tumor cells and pulmonary and hepatic fibrosis and causes injury in the same organs [8,21]. On the other hand, ACE2 affects some oxidative stress pathways via increasing Ang (1–7); if the levels of this enzyme begin to decline, the expression of NADPH oxidases 2/4 (NOX2/NOX4) increases, along with the levels of the reactive oxygen species (ROS) and loss of the stability of the mitochondrial membrane and B-cell lymphoma 2 protein (Bcl2) [22].

Additionally, the participation of microRNAs (miRNAs), such as miR-4262, related to the regulation of apoptosis mechanisms, may be involved in regulating the molecular processes related to ACE2 [23]. Through different exogenous blockers, it has been possible to identify two primary receptors for Ang (1–7), called Mas-related G-protein coupled receptor type D (MrgD) and MasR [24]. MasR is described as the central receptor associated with the axis-regulating mechanism that is encoded by the MAS1 oncogene and is characterized as a heptameric G protein-associated receptor, serving as a starting point in various signaling cascades that promote the phosphorylation of second messengers such as phosphoinositol-3-phosphate kinase (PI3K), phospholipase A2 (PLA2), and phospholipase C (PLC), among others. Another result is the decreased phosphorylation of ERK1/2, c-Jun, mitogen-activated protein kinase (MAPK), and the Smad family [25] but increased phosphorylation of endothelial nitric oxide synthase (eNOS) and cyclooxygenase 2 (COX-2), improving the vasodilation and blocking of the transcription of profibrotic and proinflammatory genes [26]. A worsening of the inflammatory processes has been documented in animals deficient in MAS1, as in arteriosclerosis, in which more significant macrophage infiltration and proinflammatory cytokine production are observed [20]. These inflammatory effects can be mitigated through the use of diminazen aceturate (DIZE), which is an ACE2 inducer that decreases the expression of interleukin-1β (IL-1β), interleukin -6 (IL-6), tumor necrosis factor-α (TNF-α), and monocyte chemoattractant protein-1 (MCP-1) [8]. Such a decrease in the inflammatory marker expression is mainly associated with the blockade of the nuclear factor kappa-light-chain-enhancer of activated B cells (NF-κB) signaling pathway, a process described at the liver, lung, and kidney levels in different study models treated with ACE2 inducers and exogenous ACE2/Ang (1–7) therapy (Figure 4) [27].

Furthermore, increases in aldosterone can block MasR expression but not that of ACE2 or Ang (1–7). Spironolactone, an aldosterone antagonist, has been shown to have effects on this receptor, in addition to promoting a balance towards the ACE2/Ang (1–7) axis [28].

## 4. ACE2 in the Pathophysiology

Most studies on ACE2 to date have focused on heart disease, particularly on the RAS balance, participating in the regulation of blood pressure and flow, electrolyte balance, and vascular resistance [25]. The main molecules involved in the biological function of ACE2 are angiotensin I (Ang I) and Ang II, the regulators of blood pressure and the release of aldosterone from the adrenal cortex, thus promoting the reabsorption of sodium. Similarly, the ACE2 levels, increased under stress, downregulate the local inflammation-promoting cellular homeostasis [29]. Animals with ACE2 gene deletions under chemical stimuli develop significant inflammation, even liver and kidney fibrosis, demonstrating a significant homeostatic role in these organs (Figure 5) [30].

The importance and impact of the nonclassical RAS axis in different diseases regarding the main organs that express the enzyme are described below.

### 4.1. Heart

The primary function of ACE2 is the conversion of Ang-II to Ang (1–7), a heptameric molecule associated with the improvement of cardiovascular diseases [22], as well as a decrease in the expression of proinflammatory cytokines like IL-1β [8,27,31]. The Ang II/ACE/AT1R axis molecules are increased in animals conditioned to heart failure, but Ang (1–7)/ACE2 is decreased [3]. This mechanism has also been described in the processes of remodeling, hypertrophy, and ventricular fibrosis in murine models [32]. These effects have been reduced by exogenous ACE2 injection or viral transfection, which improves the conditions of the animals and changes the patterns of protein production in favor of ACE2/Ang (1–7) [33]. Furthermore, plasma ACE2 measurements are a predictive biomarker for the loss of myocardial function and increased cardiac fibrosis [34].

### 4.2. Kidneys

Heart conditions reverberate at the renal level, with renal hypertension, renal ischemia, and fibrosis being the most common [25]. Based on studies in animal models, different pharmacological therapies have been characterized that focus on the balance of the RAS axis. For example, recombinant Ang (1–7) has been shown to alleviate the progression of nephritis through decreases in collagen, fibronectin, and actin [25], and exogenous ACE2 has been used to drive the balance toward Ang (1–7) [35]. The combined use of vitamins and ions with therapies improves renal hypertension [36].

### 4.3. Nervous System

One of the main ways in which the protective effect of ACE2 has been studied is regarding Alzheimer’s disease [37]. The importance of ACE2 in preventing damage to β amyloid (Aβ) [8], a protein that has been associated with progressive disease damage, has been demonstrated in animal models. This effect has been studied in rats conditioned to this disease in which treatments with DIZE stimulate the expression of ACE2 and AT1R blockers have been applied. In response, the RAS nonclassical axis is upregulated, inflammation is reduced, and cognitive abilities in the early stages (2 weeks to 3 weeks of age) are improved; the worsening of older animals (10–13 weeks of age) is also prevented [8,38].

Exogenous Ang (1–7) and ACE2 treatments in arterial occlusion, cerebral ischemia, and cerebral hypertension also lessen the inflammatory environment and improve the brain conditions in animal models [20]. The aforementioned pharmacological effects are enhanced in combined therapy with vitamin D and are likely associated with the polarization of glial cells [39]. The increased expression of ACE2 mediated by a disintegrin and metalloprotease 17 (ADAM17) promotes the polarization of M1 to M2, reducing the inflammation and oxidative stress [4].

### 4.4. Liver

In liver lesions, ACE2 is increased to augment Ang (1–7) and recruit hepatocytes from the biliary tube epithelium to heal wounds [40]. Its use in alleviating fibrosis [41], oxidative stress, and the autophagy of hepatocytes has been proven in vitro and animal models with fibrosis and liver lesions using exogenous ACE2 as a stimulator [42]. Another model in which the beneficial effects of ACE2 have been observed is nonalcoholic cirrhosis, in which an increase in the ACE2/Ang (1–7) levels has been reported concerning the classic axis, promoting an increase in the blood flow and a decrease in inflammatory markers [43]. Similar to other conditions, viral vectors carrying the ACE2 gene have been applied in animal models of sclerosing cholangitis, resulting in decreased fibrosis and liver damage [41].

### 4.5. Cancer

ACE2 has been used as a follow-up marker in different cases of cancer. Zhan Q. and collaborators showed that a decrease in the ACE2 levels predominated; however, patients with stable levels had a better prognosis and relapse-free survival [21]. In contrast, the study by Nayaran S. and collaborators described an increase in the ratio of ACE to ACE2 in papillary thyroid carcinoma and large tumor masses [44].

### 4.6. Other Conditions

The generation of animal models and human cell studies has indicated that the ACE2/Ang (1–7)/MasR axis is widely related to type 2 diabetes mellitus, insulin resistance, and the associated neuropathic pain [45].

## 5. ACE2 in Lung Conditions

The lungs are one of the main organs in which ACE2 is expressed; this has led different groups to investigate the participation of this enzyme in various lung diseases. In Acute lung injury (ALI)-induced murine models by hyperoxia and lipopolysaccharide (LPS), an overexpression of the ACE-Ang II-AT1R axis was observed. Injecting exogenous ACE2 or using stimulators of the nonclassical axis of RAS improved wounds with a decrease in inflammatory markers (TNF-α and IL-1β) [46]. The anti-inflammatory effect mainly occurred by inhibiting the NF-κB pathway, increasing Ang (1–7), and regulating the autophagy and apoptosis processes [47].

ACE2 improved the lung functions after the ALI induction, likely regulating the permeability of pulmonary blood vessels by antagonizing vascular endothelial growth factor A (VEGF-A), controlling the blood flow [48]. ACE2-depleted animals have higher levels of MMPs, leading to an imbalance between the metalloproteinases (MMPs) and tissue metalloproteinase inhibitors (TIMPs) and more significant inflammation and remodeling [30,49], but on the other hand, they exhibit fewer lesions induced by PM2.5 (particulate matter with a mean aerodynamic diameter less than 2.5 μm) compared with those without modifications [49].

After exogenous ACE2 is administered to lung tissues, there is a decrease in the inflammatory markers, fibrosis, and lung remodeling in the murine model of bleomycin-induced pulmonary fibrosis (FP) [50]. The protective effects of ACE2 are potentiated in combination therapies with mesenchymal umbilical cord stem cells, all of which have only been tested in mice [51].

In silicosis, there is also an increase in ACE, Ang II, and AT1R and a decrease in the ACE2, Ang (1–7), and MasR levels [52]. When the tissues and lung cells affected are stimulated with Ang II, an increase in the components of the nonclassical RAS axis occurs, mainly in fibroblasts; additionally, the serum levels of Ang II decrease [53].

In pulmonary arterial hypertension (PAH), low levels of ACE2 and Ang (1–7) have been reported [54]. In animal models of hypoxia-induced lung lesions, increases in Ang (1–7) and an inversely proportional relationship with the Ang II levels have been found [55]. ACE2 also plays a protective role against the cellular apoptosis events in pulmonary embolism [56].

Some pharmacological therapies combined with vitamins [57] have successfully enhanced the expression of the nonclassical axis of RAS; cells with genetic modifications in the ACE2 promoter region have also been produced, significantly increasing the expression of the protein product [58]. However, these therapies have only been applied in animal models and in vitro in human and animal cells. It has also been argued that the early use of pharmacological treatments might have a better effect, at least in animal models [59].

In general, most of the research carried out thus far has focused on the involvement of ACE2 in fibrosis and lung inflammation processes. There are also data on its importance in viral infections. In the respiratory syncytial virus model, ACE2-deficient animals showed more severe symptoms of infection, as well as more severe lung lesions [60]. In the cases of severe infection by members of the Paramyxoviridae family, such as influenza virus A, acute lung lesions occur, which can worsen in animals with deletions of the ACE2 gene [61].

Members of the family Coronaviridae have also been shown to have the ability to use ACE2 to infect human cells, especially SARS-CoV and NL63, and SARS-CoV has a greater capacity for infection than NL63. Furthermore, cells infected with SARS-CoV show higher viral replication than those infected with NL63. In infected cells, the amount of ACE2 in the cell membrane is decreased [62].

### SARS-CoV-2 Infection

The SARS-CoV-2 virus is an emerging virus first reported in Wuhan, China at the end of 2019. Menachery et al. hypothesized the resurgence of SARS-CoV infections and the potential risk of recombinations with other viral bat populations [63]. According to phylogenetic analyses, SARS-CoV-2 is genetically related to existing viruses in bats, and it appears to be a new variant of the existing SARS-CoV [64].

An in silico analysis has shown that SARS-CoV-2 is a variation of the SARS-CoV virus, with which it shares 75% homology in its genetic sequence [65]. Applying computational models, it has been demonstrated that this new viral strain has an affinity for ACE2 [66], with which it binds through the spike protein (S protein) via amino acids at positions Y442, L472, N479, D480, T487, and Y491; indeed, these residues seem to play a critical role in its interaction with ACE2 [67].

In prospective epidemiological studies, various ACE2 sequences have been analyzed and compared in different mammals, revealing high conservation at T20, Y83, S218, A246, K353, P426, T593, N636, A714, R716, and A774, though this was not found in mice [65,67]. These studies help to identify and prevent potential intraspecies infections involving cows, cats, dogs, and pigeons, among others [65].

Crystallographic analyses of the C-terminal region and interactions with ACE2 found that the interactions are similar to those of SARS-CoV. However, the amino acids critical for their interactions could not be identified. The structure of the SARS-CoV-2 S protein is more compact than its counterpart, and two critical sites for binding to ACE2 have been recognized, causing a more stable interaction [68,69]. Nonetheless, despite the possible homology between receptors, monoclonal antibodies against SARS-CoV do not react with the new variant of the virus [69]. When assessing anti-SARS-CoV polyclonal antibodies against the new strain, an immune response against SARS-CoV-2 was initiated, which led to the hypothesis that cross-reactivity can be used as a therapy in the treatment of SARS-CoV-2 [70]. Cross-reactive antibodies are defined as antibodies that, despite being directed against a specific antigen, may have an affinity against one or more unrelated antigens. Some studies have suggested that exposure to other strains of coronavirus (e.g., human seasonal coronavirus, hCOV) can promote an immune response against SARS-CoV-2; however, the preliminary results in children showed that, although the antibodies produced against hCOV are capable of successfully binding to SARS-CoV-2, this does not induce the immune response [71]. Other investigations have focused their attention on studying this same effect induced by vaccines such as DTP, in which results have been found that could enrich the explanation of the low incidence of SARS-CoV-2 in children and young people, as well as the variability of the deaths in specific populations [72]. In silico studies have proposed potential candidate vaccines for cross-reactivity, such as the Bacillus Calmette-Guérin (BCG) vaccine [73], which can help induce the innate immune response and prevent more severe disease forms. Cross-reactivity events, plus an understanding of the mechanisms mediated by immune cells, can be a potential therapy to prevent adverse events and more severe forms of SARS-CoV-2 infections [74,75].

These data have allowed us to describe how SARS-CoV-2 uses ACE2 as an input receptor to infect human cells, blocking the protective functions of the enzyme in different organs [76]. Additionally, participation of the TMPRSS2 (Transmembrane protease serine 2) receptor during infections has been reported [14,77], along with other receptors associated with cellular apoptosis and mitochondrial survival mechanisms [78].

Due to the expression of ACE2 in the lungs, brain, liver, small and large intestines, kidneys, and heart, these organs are known to be the primary organs affected by the virus. It is worth mentioning that, although ACE2 is also expressed in the connective epithelium, SARS-CoV-2 infects predominantly through the oral/nasopharyngeal route [79].

SARS-CoV-2 infection can often be asymptomatic or lead to minor respiratory symptoms but may involve acute lung injury, direct myocardial damage, hypoxia, hypotension, an altered inflammatory status, fever, respiratory distress syndrome, septic shock, metabolic acidosis, bleeding dysfunction, coagulation, nausea, vomiting, and diarrhea [80].

Based on the current observations, the groups with the worst prognoses are usually men; older adults (>60 years); and people with comorbidities (hypertension, heart conditions, and/or diabetes) [81]. The most prevalent comorbidities associated with SARS-CoV-2 include hypertension (21.1%), diabetes (9.7%), cardiovascular diseases (8.4%), and respiratory system disease (1.5%); when compared between severe and non-severe patients, the pooled odds ratio of hypertension, respiratory system disease, and cardiovascular disease were 2.36, 2.46, and 3.42, respectively [82]. Additionally, the ACE2 genetic variants are associated with an earlier penetrance and more severe hypertension and more severe outcomes of COVID-19 in obese, smoking males [83].

Tobacco smoking and the use of electronic cigarettes have also been shown to increase the risk of presenting more severe symptoms of the disease, because they induce an overexpression of ACE2 [84,85]. The results in the investigation between ACE2 and SARS-CoV-2 are heterogeneous; experimental analyses show that ACE2 expression is increased in the epithelial lung cells of smokers [86,87]. Interestingly, the statistics are very different for children who have a better prognosis; the infections are mild to moderate, and mortality is fortunately almost nonexistent.

A recent in silico analysis of interactive networks of genes and proteins co-expressed with ACE2 was able to identify potential drugs that might treat the different symptoms associated with SARS-CoV-2 infection, including nimesulide, fluticasone propionate, thiabendazole, photopyrin, didanosine, and flutamide [88]. However, the use of classic RAS axis blockers may also be beneficial [89].

Given the importance of ACE2 as a SARS-CoV-2 receptor, a general concern was generated after finding hypertension a risk factor for increased complications and death risks after COVID-19. A hypothesis grew that an increase in COVID-19 risk was due to the use of ACE blockers as antihypertensive drugs or to modify the ACE2 receptors occurring, for example, in diabetes. However, antihypertensives are ACE inhibitors and do not inhibit ACE2, and angiotensin II type 1 receptor blocker, another group of commonly used antihypertensive, do not consistently upregulate ACE2 receptors, and it is now being considered that those drugs do not need to be suspended [90].

## 6. Genetic Aspects Related to ACE2

Genome-Wide Association Studies (GWAS) have helped to describe the potential risk signals near to CXCR6 and the ABO blood group locus [91], as well as in the HLA loci, TMEM189-UBE2V1, and TMPRSS2 [92]. On the other hand, GWAS studies have shown a genetic heterogeneity in the factor associated with the risk of suffering COVID-19; some studies associated different loci—for example, in chromosomes 2, 7, 16, and 17—where most of the associations are related to severe forms of infection or biologically related to inflammation mechanisms [93].

Various genetic association studies have been carried out regarding ACE2 polymorphisms and different diseases, mainly hypertension and other cardiac disorders [94]. Through a meta-analysis, the genetic variants for the RAS axis were identified as mainly insertions in ACE2 and single-nucleotide polymorphisms in AGT (rs699) and AGTR1 (rs5186), which are associated with a lower risk of chronic kidney disease [95].

Nonetheless, a recent comparative genetic analysis of ACE2 as a SARS-CoV-2 receptor failed to identify the existence of genetic variants in the ACE2 gene that confer a resistance to the binding of the coronavirus S protein in different populations [96]. In silico models of ACE2 have analyzed possible genetic variants that may affect its interaction with the SARS-CoV-2 S protein. Indeed, rs73635825 (S19P) and rs143936283 (E329G) were shown to interfere with the enzyme’s interaction with the viral protein [97].

## 7. Conclusions

ACE2 is an enzyme with a wide distribution in the organs and cells of the human body; its primary function is regulating the blood pressure, decreasing inflammation and fibrosis, and relieving the damage induced in different organs. The SARS-CoV-2 pandemic has made it necessary to increase research to find possible therapeutic targets that can prevent infections or avoid more severe forms of the infection. ACE2 is a potential therapeutic target, as it is one of the main entry points for SARS-CoV-2. However, research has been carried out on knowing its genetic variability and molecular protein structure, and it is necessary to discover how to apply this information in clinical and therapeutic settings. The genetic component has shown significant heterogeneity, making it difficult to carry out replication studies. However, the genetic association studies have included SNPs in candidate genes; some of the most significant results are related to multiple loci, giving new insights into the study of SARS-CoV-2.

In vitro experiments have been performed for predicting possible peptide sequences capable of blocking the binding sites between ACE2 and the spike protein of SARS-CoV-2. These hypotheses need to be tested with experimental models. Additionally, clinical trials with drugs that may intervene in the interactions, as mentioned above, should be launched.

## Figures and Tables

**Figure 1 pharmaceuticals-14-00805-f001:**
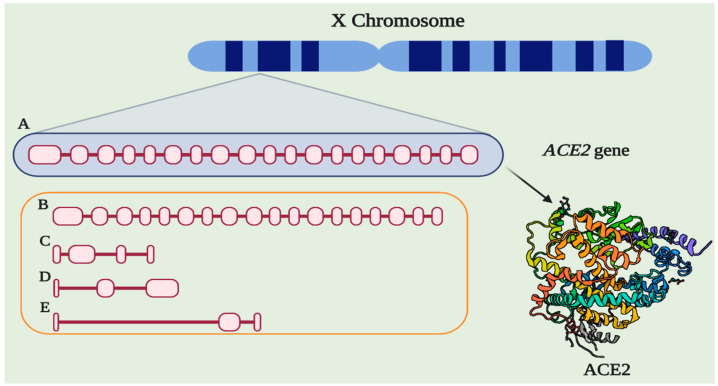
The structure of ACE2. (A) Gene ACE2 conformed by 19 exons and 18 introns (3507 bp). (B) Alternative functional transcript without the last exon (3339). (C) Alternative transcript without the protein product (4 exons, 998 bp). (D) Alternative transcript without the protein product (3 exons, 786 bp). (E) Alternative transcript without the protein product (3 exons, 599 bp). Protein structure-based in PDB 1R42.

**Figure 2 pharmaceuticals-14-00805-f002:**
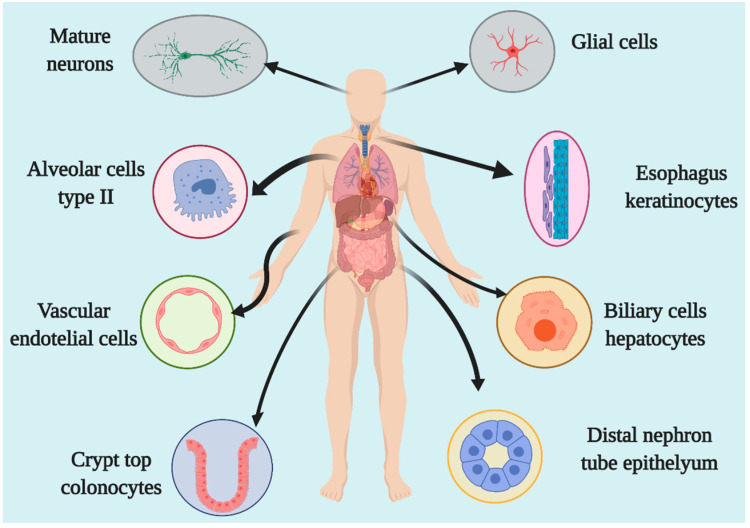
The organs and cells expressing ACE2.

**Figure 3 pharmaceuticals-14-00805-f003:**
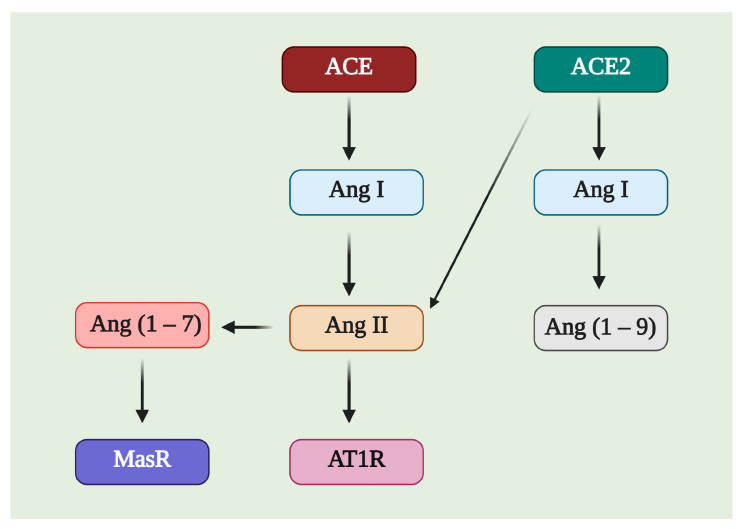
Image of the possible substrates and products of Angiotensin-Converting Enzyme 2 (ACE2). ACE2 can affect Ang I and Ang II, producing Ang (1–9) and Ang (1–7), respectively. Ang (1–7) is a molecule with affinity for the Mas receptor (MasR), while Ang II is related to angiotensin II type 1 receptor (AT1R).

**Figure 4 pharmaceuticals-14-00805-f004:**
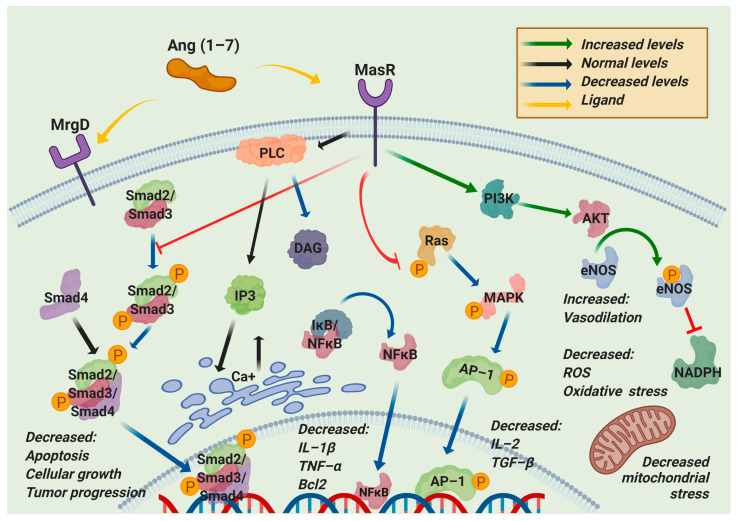
Regulated pathways of the nonclassical RAS axis. Angiotensin (1–7) (Ang (1–7)) can bind to Mas-related G-protein coupled receptor type D (MrgD) or Mas receptor (MasR). This interaction blocks the phosphorylation of the components of the two main pathways, Ras/Mitogen-activated protein kinase (MAPK) and Smad, resulting in the downregulation of apoptosis, cellular growth, tumor progression, inflammation, oxidative stress, reactive oxygen species, and mitochondrial stress. On the other hand, this interaction upregulates phosphatidylinositol-3-kinase (PI3K) to increase endothelial nitric oxide synthase (eNOS) phosphorylation, blocking the nicotinamide adenine dinucleotide phosphate reduced (NADPH) activity, decreasing the oxidative stress, and increasing the vasodilation. PLC: Phospholipase C; DAG: diacylglycerol; IP3: Inositol trisphosphate; IkB: inhibitor of nuclear factor kappa; NF-κB: nuclear factor kappa-light-chain-enhancer of activated B cells; IL-1β: interleukin 1 beta; TNF-α: Tumor necrosis factor alpha; Bcl2: B-cell lymphoma 2; Ras: Rat sarcoma virus; AP-1: activator protein 1; AKT1: AKT serine/threonine kinase 1; ROS: reactive oxygen species.

**Figure 5 pharmaceuticals-14-00805-f005:**
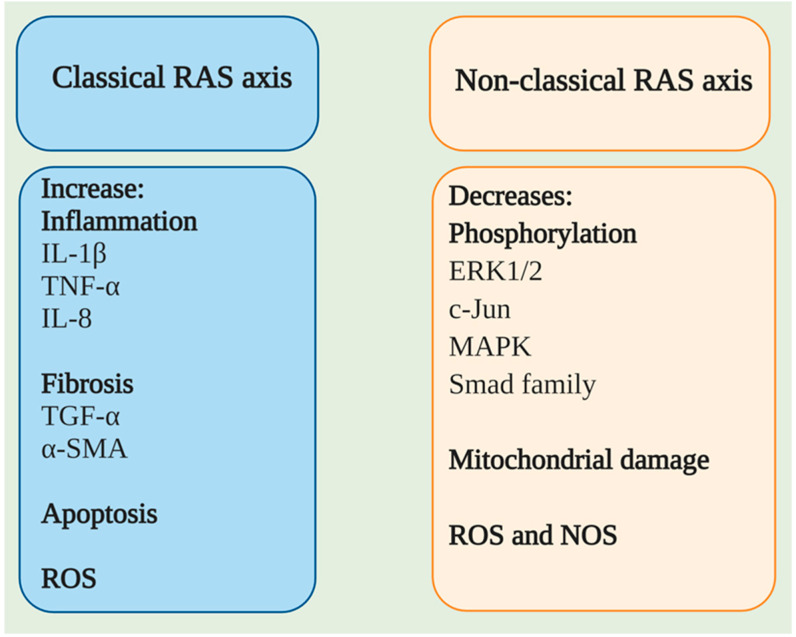
Classical and nonclassical renin-angiotensin (RAS) axes. Biological effects of the classical (ACE/Ang II/AT1R) and nonclassical (ACE2/Ang (1–7)/MasR) axes. IL-1β: interleukin 1 beta; TNF-α: Tumor necrosis factor alpha; Bcl2: B-cell lymphoma 2; IL-8: interleukin 8, TGF-β: Transforming growth factor beta; α-SMA: alpha smooth muscle actin; ROS: Reactive oxygen species; ERK1/2: extracellular signal-regulated protein kinase 1/2; NOS: Nitric oxide synthase.

## Data Availability

Data sharing not applicable.

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
