# Peer review of "Angiotensin-Converting Enzyme 2 (ACE2) in the Context of Respiratory Diseases and Its Importance in Severe Acute Respiratory Syndrome Coronavirus 2 (SARS-CoV-2) Infection"

_pharmaceuticals, 2021, doi:10.3390/ph14080805_

Round 1
Reviewer 1 Report
Thank you for giving me the opportunity to read and comment a report “ACE2 in the context of respiratory diseases and its importance 2 in SARS-CoV-2 infection”, by Ambrocio-Ortiz et al.
In the reviewed manuscript, the role of ACE2 in respiratory diseases and its importance with regard to SARS-CoV-2 is reported. The authors made an in-depth review about ACE2, specifically, location and expression, pathophysiology and its relation with SARS-CoV-2 development.
This paper is well written, correctly structured with a suitable research concept and definitely it is of relevance to readers of the journal. However, some suggested minor changes are included in the comments given below.
- It would be convenient to define some abbreviations such as IL, VEGFR2, MEK, among others
- The acronym ACE2 appears type in several diverse ways throughout the report, for example, ACE2 or Ace2. Please unify the acronym writing.
- Please, explain acronyms the first time you use in the manuscript.
- The review carried out in this manuscript is concrete and concise, but the conclusions are too vague and refer to future studies. In short, they do not fully respond to the established objectives.
- The bibliography format is incorrect. It does not correspond to the journal recommendations.
Author Response
I appreciate the time spent reading and reviewing the submitted article, as well as the comments made. Below you can find the annotations regarding the comments that you have kindly provided us:
Reviewer 1
- It would be convenient to define some abbreviations such as IL, VEGFR2, MEK, among others
Thank you for your review and comments; we check all the abbreviations of the article and add the meaning of each one.
- The acronym ACE2 appears type in several diverse ways throughout the report, for example, ACE2 or Ace2. Please unify the acronym writing.
Thank you for your observation. The article describes some experiments and discoveries made in animals; ACE2 refers to the human protein, while Ace2 refers to the homolog in animals.
- Please, explain acronyms the first time you use in the manuscript.
Thank you for your review and comments. As we answer in the first observation, we check all the abbreviations of the article and add the meaning of each one.
- The review carried out in this manuscript is concrete and concise, but the conclusions are too vague and refer to future studies. In short, they do not fully respond to the established objectives.
ACE2 is an enzyme with a wide distribution in the organs and cells of the human body; its primary function is regulating blood pressure, decreasing inflammation and fibrosis, and relieving damage induced in different organs. The SARS-CoV-2 pandemic has made it necessary to increase research to find possible therapeutic targets that can prevent infections or avoid more severe forms of infection. ACE2 is a potential therapeutic target as it is one of the main entry points for SARS-CoV-2; However, research has been carried out focused on knowing its genetic variability and molecular protein structure, it is necessary to delve into discovering how to apply this information in clinical and therapeutic settings.
- The bibliography format is incorrect. It does not correspond to the journal recommendations.
We appreciate the observation made; we have reviewed the citations included and have made the pertinent corrections to leave them in the format requested by the journal.
Reviewer 2 Report
This is a well written review, nice to read and easy to understand,
that focuses on the importance of ACE-2 functions with regard to SARS-CoV2.
Few comments:
1) As the main topic of this review is the role of ACE-2 in the context of SARS-CoV-2 infection, I would expand this part, deepening a few aspects that are still debated in literature (e.g. the association between smoking or gender and ACE-2 expression).
2) As ACE-2 is so important in SARS-CoV-2 infection, allowing the viral entry, has its expression been investigated specifically in COVID-19 patients?
3) Page 2, lines 67-71 (The primary cells...oral cavity also expresses this enzyme"). It seems to repeat what already said in the previous sentence.
4) Page 2, lines 77: Has a reference been skipped?
5) Few typos in the text.
Author Response
Reviewer 2
- As the main topic of this review is the role of ACE-2 in the context of SARS-CoV-2 infection, I would expand this part, deepening a few aspects that are still debated in literature (e.g. the association between smoking or gender and ACE-2 expression).
Thank you for your review and comments; the most prevalent comorbidities associated with SARS-CoV-2 infection include hypertension (21.1%), diabetes (9.7%), and cardiovascular diseases; when odds ratios were calculated, hypertension (2.36), other respiratory diseases (2.46) and cardiovascular diseases (3.42) (PMID: 32173574). When the genetic component is included in the studies, the risk of suffering severe cases of COVID-19 increases, and even with other comorbidities (PMID: 33386398), now this information was included in lines 327-333.
The results in the investigation between ACE2 and SARS-CoV-2 are heterogeneous, and experimental analyses show that ACE2 expression is increased in epithelial lung cells of smokers (PMID: 32425701, 32991738), now this information was included in lines 335-338 .
- As ACE-2 is so important in SARS-CoV-2 infection, allowing the viral entry, has its expression been investigated specifically in COVID-19 patients?
That’s an excellent question; studies about ACE2 and other potential receptors have been studied in different biological samples. The expression of ACE2 is increased in patients infected with SARS-CoV-2 in the airway epithelium and immunological cells, but not in other epithelia (PMID: 32303424, 32496587, 32374427). In some studies, the differences are not significant. In the analysis of patients infected with SARS-CoV-2, including environmental variables like smoking or other lung diseases (asthma, COPD), the expression of ACE2 is increased significantly, and, interestingly, this difference increase in severe COVID-19 (PMID: 32496587).
Now we have added some of these ideas in lines 80-86 and 336-338.
- Page 2, lines 67-71 (The primary cells...oral cavity also expresses this enzyme"). It seems to repeat what already said in the previous sentence.
Thanks for your observation; we resume the sentence and correct the expression of the subsequent paragraphs.
- Page 2, lines 77: Has a reference been skipped?
Thanks for your advice; we check the references list; these lines should not be included in the final text.
- Few typos in the text.
We checked the typos and corrected them.
Reviewer 3 Report
Dear Authors,
I’ve appreciated your paper. In my opinion your review could help to better understand the role of ACE2 , especially in clinical manifestations of SARS Covid-19.
Your figures are very clear and useful, with a special mention to n° 4 and 5.
Above are listed few suggestions before editing :
- In opening Chapter 3 (pag. 3 line 81) you could underline that ACE2 is a member of the counter-regulatory axis of RAS, and its main role is to degrade the pro-hypertrophic and profibrotic peptide, angiotensin II, limiting the adverse effects of angiotensin II. Its activation has been shown to offer some protective effect against fibrosis and inflammation process.
- When you discuss about relationship between SARS - Covid 19 and ACE 2 (pag 9 line 280) you should better explain and define “cross-reactivity” as therapeutic strategy.
- In your conclusion I’d like to read some topics for future research concerning ACE 2 and SARS Covid 19 relationship.
Best regards
Author Response
I appreciate the time spent reading and reviewing the submitted article, as well as the comments made. Below you can find the annotations regarding the comments that you have kindly provided us:
- In opening Chapter 3 (pag. 3 line 81) you could underline that ACE2 is a member of the counter-regulatory axis of RAS, and its main role is to degrade the pro-hypertrophic and profibrotic peptide, angiotensin II, limiting the adverse effects of angiotensin II. Its activation has been shown to offer some protective effect against fibrosis and inflammation process.
We appreciate your advice, and we have taken into account this information that you have not provided to adapt it in our writing.
- When you discuss about relationship between SARS - Covid 19 and ACE 2 (pag 9 line 280) you should better explain and define “cross-reactivity” as therapeutic strategy.
Cross-reactive antibodies are defined as antibodies, which, despite being directed against a specific antigen, may have affinity against one or more unrelated antigens. Some studies have suggested that exposure to other strains of coronavirus (e.g., human seasonal coronavirus, hCOV), can promote an immune response against SARS-CoV-19; however, preliminary results in children show that although the antibodies produced against hCOV are capable of successfully binding to SARS-CoV-2, this does not induce the immune response (Humoral immunity to SARS-CoV-2 and seasonal coronaviruses in children and adults in north-eastern France). Other investigations have shown their attention to studying this same effect induced by vaccines such as DTP, in which results have been found that could enrich the explanation of the low incidence of SARS-CoV-2 in children and young people, as well as the variability deaths in certain populations. (Potential Cross-Reactive Immunity to SARS-CoV-2 from Common Human Pathogens and Vaccines). In silico studies have proposed potential candidate vaccines for cross-reactivity, such as the Bacillus Calmette – Guérin (BCG) vaccine (Identification of similar epitopes between severe acute respiratory syndrome coronavirus-2 and Bacillus Calmette – Guérin: potential for cross-reactive adaptive immunity) that can help induce the innate immune response and prevent more severe forms of the disease. Cross-reactivity events, plus an understanding of the mechanisms mediated by immune cells, can be a potential therapy in the prevention of adverse events and more severe forms of SARS-CoV-2 infection (Potential CD8+ T Cell Cross-Reactivity Against SARS-CoV-2 Conferred by Other Coronavirus Strains, CD8 T cell epitope generation toward the continually mutating SARS-CoV-2 spike protein in genetically diverse human population: Implications for disease control and prevention). Cross-reactivity events, plus an understanding of the mechanisms mediated by immune cells, can be a potential therapy in preventing adverse events and more severe forms of SARS-CoV-2 infection.
- In your conclusion I’d like to read some topics for future research concerning ACE 2 and SARS Covid 19 relationship.
In vitro experiments have been performed predicting possible peptide sequences capable of blocking the binding sites between ACE2 and the spike protein of SARS-CoV-2; these hypotheses need to be tested with experimental models. Also, clinical trials with drugs that may intervene in the interaction above should be launched.